# Synthesis and Anticancer Activity of 1,3,4-Thiadiazoles with 3-Methoxyphenyl Substituent

**DOI:** 10.3390/molecules27206977

**Published:** 2022-10-17

**Authors:** Sara Janowska, Dmytro Khylyuk, Agnieszka Gornowicz, Anna Bielawska, Michał Janowski, Robert Czarnomysy, Krzysztof Bielawski, Monika Wujec

**Affiliations:** 1Department of Organic Chemistry, Faculty of Pharmacy, Medical University, 4a Chodzki Str., 20-093 Lublin, Poland; 2Department of Biotechnology, Faculty of Pharmacy, Medical University of Bialystok, Kilinskiego 1 Street, 15-089 Bialystok, Poland; 3Department of Synthesis and Technology of Drugs, Faculty of Pharmacy, Medical University of Bialystok, Kilinskiego 1 Street, 15-089 Bialystok, Poland

**Keywords:** thiosemicarbazides, 1,3,4-thiadiazoles, anticancer activity, breast cancer, mechanism of action

## Abstract

Based on the results of previous work, we designed and synthesized 1,3,4-thiadiazole derivatives. The cytotoxic activity of the obtained compounds was then determined in biological studies using MCF-7 and MDA-MB-231 breast cancer cells and a normal cell line (fibroblasts). The results showed that all compounds displayed weak anticancer activity towards two breast cancer lines: an estrogen-dependent cell line (MCF-7) and an estrogen-independent cell line (MDA-MB-231). The compound most active towards MCF-7 breast cancer cells was SCT-4, which decreased DNA biosynthesis to 70% ± 3 at 100 µM. The mechanism of the anticancer action of 1,3,4-thiadiazole was also investigated. We choose a set of the most investigated proteins, which are attractive anticancer targets. In silico studies demonstrated a possible multitarget mode of action for the synthesized compounds but the most likely mechanism of action for the new compounds is connected with the activity of caspase 8.

## 1. Introduction

The search for new methods of pharmacotherapy of oncological diseases is currently among the most important challenges of modern medicine. In 2018, cancer was diagnosed in 18.1 million people [1]. In 2020, 19.3 million new cancer cases in patients and nearly 10.0 million related deaths were reported worldwide [2]. The number of cancer cases is growing every year. According to data from the World Health Organization (WHO), it is estimated that this number will exceed 29 million cases per year by 2040 [2]. Breast cancer is currently the most commonly diagnosed cancer in the world. According to current statistics, 2.26 million cases of breast cancer were detected in 2020. At the same time, this disease is responsible for the highest number of deaths from oncological causes in women on a global scale [3]. In 2020, breast cancer killed nearly 685,000 women worldwide [4].

Different biological types and stages of development of breast cancer require a different therapeutic approach [5]. Therefore, research enabling a better understanding of the biology and molecular basis of the disease process is key to developing better treatments for breast cancer [5]. Currently available conventional methods of breast cancer therapy include local treatment, such as surgical interventions and radiotherapy, and systemic treatment, such as chemotherapy, hormone therapy, targeted therapy, or immunotherapy [5]. The high mortality rate associated with this disease indicates the need for further search for new forms of pharmacotherapy. Among the most important problems in the pharmacological treatment of cancers, including breast cancer, is rapidly developing drug resistance [5]. With subsequent unsuccessful attempts at chemotherapeutic treatment, the sensitivity of cancer cells to administered drugs decreases. Current standard drugs and chemopreventive agents, such as tamoxifen, used in breast cancer most often work through the estrogen receptor (ER) mechanism. Therefore, these pharmaceuticals are ineffective against estrogen-independent diseases. Consequently, searches for new therapeutic agents, the activity of which would be independent of the ER status, are justified. Such drugs would be appropriate in cases of estrogen-independent breast cancer [6].

In recent years, molecules containing five-membered heterocycles have become an object of increasing interest in the context of designing anticancer compounds. One of the promising structures is the 1,3,4-thiadiazole system. The bioactive properties of thiadiazole are influenced by the fact that its heterocyclic system is a bioisoster of pyrimidine, which in turn is the backbone of three nucleobases [7]. For this reason, 1,3,4-thiadiazole derivatives have the ability to interfere with processes related to DNA replication. As a result, the derivatives of this heterocyclic compound have the potential to inhibit both tumor and bacterial cell division [8].

There are many research reports on the antitumor therapeutic potential of 1,3,4-thiadiazole derivatives. The reported compounds include thiadiazole systems fused with other rings and simple 2,5-disubstituted rings. Most of the published research results indicate the therapeutic potential of thiadiazole molecules containing a simple unfused ring. Some of the tested compounds showed promising antitumor activity in in vitro biological tests, exceeding the reference drugs in the tests. Many studies have also demonstrated the activity of compounds from this group against breast cancer cell lines [9,10,11,12,13,14,15,16,17,18,19,20,21,22,23,24,25].

Based on our previous work [26] in which we designed and synthesized a series of 1,3,4-thiadiazoles and then tested their antitumor activity against the MDA-MB-231 and MCF-7 cell lines, we developed a new series of compounds. The new series is a modification of the structures of the most active compounds from the previous work [26]. To check whether the activity of 1,3,4-thiadiazole is more influenced by the substituent attached to the ring with the mobile amino group, which allows its movement, or by the substituent rigidly attached directly to the heterocyclic ring, we swapped the substituents in the new molecules. Next, we tested the cytotoxic activity of the newly synthesized compounds against the MDA-MB-231 non-estrogen-dependent breast cancer and MCF-7 estrogen-dependent breast cancer cell lines. Additionally, we conducted in silico tests in order to explain the molecular basis of the activity of the developed compounds, examining their activity against caspase 3, caspase 7, caspase 8, Bcl-xl, Bcl2 and BAX. Flow cytometry analysis of MCF-7 and MDA-MB-231 breast cancer cells after 24 h incubation with two tested compounds (SCT4 and SCT5) was also performed to explain their effect on programmed cell death.

## 2. Results and Discussion

### 2.1. Chemistry

The synthesis of thiosemicarbazides and the 1,3,4-thiadiazole derivatives was carried out according to the synthetic route presented in Figure 1. Synthesis pathway A shows the reactions leading to the synthesis of 1,3,4-thiadiazoles having a 3-methoxyphenyl moiety in the 5-position, while the synthesis pathway B shows a method for the preparation of 1,3,4-thiadiazoles containing a 3-methoxyphenyl moiety attached to a heterocyclic ring at the 2-position. The synthesis of compounds obtained by route B was presented in an earlier article [26]. SCT7′ molecule has two 3-methoxyphenyl moieties symmetrically located both in the 2 and 5 position.

The reactions of 3-methoxyphenyl isothiocyanate and appropriate hydrazides lead to 1-*R*-4-(3-methoxyphenyl)thiosemicarbazides with good yields (46–99%). The obtained compounds were cyclized in concentrated sulfuric acid to the corresponding 1,3,4-thiadiazoles, which were obtained with a lower yield of 32–53%. The structures of the new compounds were determined using IR, ^1^H NMR, ^13^C NMR spectroscopy, and elemental analysis.

The ^1^H NMR spectra showed the chemical shifts of the protons related to atoms N1, N2, and N4 which confirmed the formation of the thiosemicarbazide scaffold. The protons were observed between 9.73 and 9.86 and 10.39 and 10.80 ppm as two singlets. The ^1^H NMR spectra confirmed the formation of thiadiazole derivatives. The N1, N2, and N4 protons of the thiosemicarbazide were not detected in the 1,3,4-thiadiazole compounds. Instead, N-H peaks of the amino group of cyclic compounds were observed between 10.47 and 10.69 ppm. The ^13^C NMR spectra of the carbons of all compounds showed the carbon signals due to the resonance of the methoxy group in the range 55.50–55.58 ppm. Signals for other carbons of synthesized molecules were observed at expected values of chemical shift.

Unfortunately, it was not possible to obtain the 1,3,4-thiadiazole derivative with a 2-trifluoromethylphenyl substituent in position 5. The change of the reaction conditions consisted of extending the time and watering the reactants did not lead to the desired compound. The synthesis with microwaves at elevated pressure was also unsuccessful.

The ^1^H NMR and ^13^C NMR spectra for all new compounds are presented in Appendix A.

### 2.2. Biological Investigations

In previous work, we described the anticancer activity of 1,3,4-thiadiazole derivatives with 3-methoxyphenylamino substituent in position 5. The strongest anticancer activity against MCF-7 was exhibited by the compound with 2-trifluoromethyphenylamino substituent in position 2. A moderate inhibitory effect on the survival of MCF-7 cells was demonstrated by two compounds with 3-methoxyphenylamino and 4-methoxyphenylamino substituents. The moderate anticancer activity against this cell line was exhibited by compound with 4-chlorophenylamino ring.

Here, we checked the cytotoxic properties of newly synthesized compounds (SCT-1, SCT-2, SCT-4, SCT-5, and SCT-6) towards MCF-7 and MDA-MB-231 breast cancer cells as well as human skin fibroblasts after 24 h incubation.

We did not observe the high cytotoxic activity of the tested compounds against breast cancer cells. SCT-4 was the most active compound in the analyzed group, and we proved that SCT-4 reduced cell viability of MCF-7 breast cancer cells to 74% ± 3 at the dose of 100 µM (Table 1).

We demonstrated that SCT-5 was most active against MDA-MB-231 breast cancer cells. SCT-5 decreased the viability of analyzed breast cancer cells to 75% ± 2 at 100 µM concentration (Table 1).

All analyzed compounds did not exert cytotoxic potential against human skin fibroblasts. At 100 µM concentration, the viability of fibroblasts was higher than 90% (Table 1). The highest viability of human skin fibroblasts was detected after 24 h of incubation with SCT2. We proved that 96% ± 1 of the cells were alive.

Comparing the results obtained in the previous work with the results of the presented research (Figure 1, Figure 2 and Figure 3), it can be concluded that 1,3,4-thiadiazole derivatives with a 3-methoxyphenyl substituent in the 5-position are definitely more active than those having the same substituent at the nitrogen atom in the 2-position of the 1,3,4-thiadiazoles. Activity in this class of compounds is determined by the substituent derived from the starting hydrazide. The graphs (Figure 1, Figure 2 and Figure 3) show a comparison of the survival rate of cells from two series of the experiment which is the test carried out for the compounds described in this paper (SCT) and their mirror-image analogs with inversely located substituents (SCT), described in the previous work [26]. The cytotoxic activity test for both lots was run under the same conditions. Tumor cells and normal fibroblasts were exposed to the test compounds at various concentrations, including 100 µM, for 24 h. The graphs were prepared in order to compare the activity of thiadiazoles with a 3-methoxyphenyl moiety in position 2 with compounds with this moiety in position 5. This summary allows to illustrate the significant influence of the 3-methoxyphenyl position on the cytotoxic activity of thiadiazole derivatives, which was discussed in more detail in this work.

The collected data suggest that the cytotoxic activity of the disubstituted 1,3,4-thiadiazole phenyl derivatives is largely determined by the type and position of the substituent on the aromatic ring bonded directly to the heterocycle. The substituents located on the aromatic ring connected to the heterocyclic system through the amino group are of less importance for the antitumor potential of the compound. The analysis of the obtained results also shows the importance of the presence of a phenyl moiety with a methoxy substituent for the antitumor activity of 1,3,4-thiadiazole derivatives. The compound obtained by our team, containing two 3-methoxyphenyl groups in the structure, showed exceptionally high activity compared to other molecules (viability of cells in 100 µM [%] 40.30 ± 2 for MCF-7 and 33.86 ± 2 for MDA-MB-231). Among the compounds from the new series, the highest activity against the MCF-7 cell line (viability of cells in 100 µM [%] = 73.56 ± 3) was demonstrated by the compound having the 4-methoxyphenyl group in position 5.

Research results confirming our conclusions can be found in the literature data. In 2011, Dalip Kumar and his colleagues synthesized a series of 2-arylamino-5-aryl-1,3,4-thiadiazoles and tested the anticancer properties of the compounds obtained. Three of them showed very high cytotoxic activity and all of them had a 3,4,5-trimethoxyphenyl group in position 5. The most active compound showed an IC_50_ = 6.6 µM against the MCF-7 breast cancer cell line [27].

At the second step of biological studies, [^3^H]-thymidine incorporation into DNA in MCF-7 and MDA-MB-231 breast cancer cells as well as human skin fibroblasts was measured after 24 h of incubation with the tested agents (SCT-1, SCT-2, SCT-4, SCT-5 and SCT-6) to see the effect of the tested compounds on DNA biosynthesis (Table 2).

We observed a rather weak antiproliferative potential of newly synthesized compounds. The most active compound towards MCF-7 breast cancer cells was SCT-4, which decreased DNA biosynthesis to 70% ± 3 at 100 µM concentration.

SCT-5 at the highest rate inhibited DNA biosynthesis in MDA-MB-231 breast cancer cells. We proved that SCT-5 decreased the analyzed process to 71% ± 3 at 100 µM concentration in the MDA-MB-231 cell line.

Similarly, in human skin fibroblasts, we did not observe the antiproliferative activity of the tested compounds. At 100 µM concentration, all compounds did not reduce [^3^H]thymidine incorporation into DNA. SCT-2 exerted the weakest antiproliferative activity against normal cells (94%). The graphs (Figure 4 and Figure 5) show the comparison of the experimental results for two series of synthesized compounds [26]. The antiproliferative activity test for both lots was run under the same conditions. MCF-7 and MDA-MB-231 tumor cells were exposed to 24 h of exposure to test compounds at various concentrations, including 100 µM. Graphs were drawn to compare the activities of thiadiazoles having a 3-methoxyphenyl moiety in position 2 with compounds having this moiety in position 5.

### 2.3. Docking Studies

For a better understanding of the mechanism of the anticancer action of 1,3,4-thiadiazole and the influences of the different substituents for its expression docking studies have been performed. We choose a set of the most investigated proteins, which are attractive anticancer targets (caspase 3-1GFW, caspase 8-3KJN, caspase 7-1SHL, Bcl-xl-2YXJ, Bcl2-2W3L, BAX-1F16). We used those models for the simulations or docking parameters as in a previous article [26] (50 runs, 300 conformational possibilities, 50 populations and 2,500,000 energy evaluations, a maximum number of 106 energy evaluations, a mutation rate of 0.02, and a crossover rate of 0.80). For caspase 7 instead of the FICA, we used the CHEMBL60190 as reported in the ligand literature [28].

Docking simulation revealed the decreased binding energy for the compounds SCT1, SCT2, and SCT4–SCT6 to most enzymes (Table 3). In particular, we observed binding energy decreases in caspase 8–SCT complexes, which confirms the theory about the relationship between the cytotoxic activity of the synthesized compounds and its influence on caspase 8. With the aim of revealing the influence of 3-methoxyphenyl substituent in 5 position of the 1,3,4-thiadiazole ring, we overlaid the structures of SCT4 and SCT4′ from previous work inside the active side of caspase 8.

As we can see under visualizations, the positions of the SCT4 and SCT4′ inside the secondary binding pocket of Caspase 8 are closed in the same way (Figure 6). Nevertheless, the 3-methoxy group of SCT4′ forms a strong short hydrogen bond with Arg260 (1.82 Å) compared to the much longer hydrogen bond with 4-methoxy group-Cys360 (2.97Å) of SCT4–caspase 8 complex. In addition, the 1,3,4-thiadiazole core of the SCT4 forms a hydrogen bond with Arg413 (1.93 Å), instead of weak lipophilic sulfur and Pi–cation interactions.

Good binding energies are observed for Bax–SCT complexes, which are quite similar to binding energy for Zinc14750348 (Table 4). That compound (also often named compound 106) demonstrated good anticancer activity in vitro and in vivo [29]. Despite the good binding energies, SCT6 interacts with BAX between α5, α6 and α9 helices, whereas Zinc14750348 occupies the carboxyl-terminal transmembrane helix binding site and binds with the site pocket in the Bax hydrophobic groove [30]. This difference may be the reason for the weak cytotoxic activity of SCTs, even with good energies of the Bax–SCT complexes.

### 2.4. Analysis of Caspase 3/7 and 8 Activity by Flow Cytometry

To confirm the results obtained by the docking studies, we analyzed the effect of the compounds tested on caspase 3/7 and 8 activity using in vitro studies. To this end, we decided to evaluate caspase 3/7 and 8 activity in MCF-7 and MDA-MB-231 breast cancer cells by flow cytometry after 24 h of treatment with tested compounds (SCT4 and SCT5, both at concentration of 100 μM). With SCT4 and SCT5 changes in caspase 3/7 activity were observed compared to control (Figure 7). Following SCT4 treatment, caspase 3/7 activation was observed in 6.1 ± 0.5% 8.0 ± 0.4% of the MCF-7 and MDA-MB-231 cell populations, respectively, while SCT5 treatment resulted in an even greater increase in the active form of caspase 3/7 to 10.3 ± 0.7% (MCF-7) and 10.8 ± 0.6% (MDA-MB-231).

Additionally, we attempted to determine by flow cytometry the activity of caspase 8 in breast cancer cells (MCF-7 and MDA-MB-231) treated (24 h) with SCT4 and SCT5 (both at concentration of 100 μM). Similar to the previous assay, tested compounds affected an increase in the active form of caspase 8 (Figure 8). The percentage of MCF-7 cells with active caspase 8 reached 5.1 ± 0.6% (SCT4) and 11.6 ± 1.4% (SCT5) of the tested cell population. For MDA-MB-231 cancer cells, the value was oscillating at 9.3 ± 0.7% and 11.2 ± 0.8% after SCT4 treatment and SCT5 treatment, respectively.

The result of caspase activation is the process of apoptosis [31]. Therefore, in order to confirm the above studies, we conducted a study of the effect of SCT4 and SCT5 compounds (24 h incubation, both at concentration of 100 μM) on the apoptosis process (Figure 9). Annexin V-FITC and Propidium iodide were the dyes used in this study. In MCF-7 cells, 12.9 ± 0.9% and 16.6 ± 1.1% of apoptotic cells were detected for SCT4 and SCT5, respectively; while in MDA-MB-231 these results were similar and amounted to 10.1 ± 0.9% (SCT4) and 13.8 ± 0.8% (SCT5). These observations indicate that the obtained results correspond well with the results of the caspase 3/7 and 8 activity tests.

## 3. Materials and Methods

### 3.1. Chemistry

#### 3.1.1. General Comments

All the substances were purchased from Sigma-Aldrich (Munich, Germany) and were used without further purification. The ^1^H and ^13^C NMR spectra were recorded on the BrukerAvance 300 (Bruker BioSpin GmbH, Rheinstetten, Germany) in DMSO-*d*_6_. IR spectra were recorded by Nicolet 6700 spectrometer (Thermo Scientific, Philadephia, PA, USA). The melting points were determined on the Stuart SMP50 melting point apparatus (Cole Parmer Ltd., Stone, UK) and are uncorrected. The purity of the compounds and the progress of the reaction were monitored by TLC (aluminum sheet 60 F254 plates (Merck Co., Kenilworth, NJ, USA). We used the solvent system CHCl_3_/EtOH (10:1, *v*/*v*). The elemental analyses were determined by a PerkinElmer 2400 series II CHNS/O analyzer (Waltham, MA, USA). Three of the obtained compounds were described by other authors (SC1, SCT1, and SCT4), but we give their physicochemical characteristics because it is absent for two compounds and it is different for one.

#### 3.1.2. Synthesis of Thiosemicarbazide Derivatives

Synthesis of SC2, SC3 and SC6

First, 0.001 mol appropriate aryl hydrazide* and 5 mL of 96% ethanol were placed in a round bottom flask. It was heated under reflux until a clear solution was obtained. An equimolar amount of the of 3-methoxyphenyl isothiocyanate was then added to the mixture. It was heated at the boiling point for 1 h. The solution was then cooled for 12 h until the product precipitated completely. The resulting solid was filtered off and washed with small portions of diethyl ether and hot water.

* 4-fluorobenzhydrazide for SC2, 2-trifluoromethylobenzhydrazide for SC3, and 4-trifluoromethylobenzhydrazide for SC6.

Synthesis of SC1, SC4 and SC5

First, 0.001 mole appropriate aryl hydrazide* was dissolved in 5 mL of 96% ethanol by heating under reflux. An equimolar amount of 3-methoxyphenyl isothiocyanate was then added to the mixture. The reaction mixture was heated until the product precipitated. Derivatives SC4 and SC5 immediately precipitated. Compound SC1 was heated for 30 min until a solid precipitated. The mixture was then cooled for 1 h until the product precipitated completely. The resulting precipitate was filtered off and washed with small portions of diethyl ether and hot water.

* 4-chlorobenzhydrazide for SC1, 4-methoxybenzhydrazide for SC4, and 3-chlorobenzhydrazide for SC5.

1-(4-Chlorobenzoyl)-4-(3-methoxyphenyl)thiosemicarbazide (SC1) [32]

CAS 901364-10-1

Yield 81% (0.33 g), m.p. 188–190 °C. Spectral data were as follows: IR (cm^−1^) KBr: 3193 (NH), 3004 (CH aliph.), 1638 (C=O), 1594 (CH arom.), 1359 (C=S), 1285 (C-O-C). ^1^H NMR (DMSO-*d*_6_) δ (ppm): 3.75 (s, 3H, CH_3_), 6.75 (d, 1H, ArH, *J* = 8.2 Hz), 7.04 (d, 1H, ArH, *J* = 8.0 Hz), 7.13 (bs, 1H, ArH), 7.24 (t, 1H, ArH, *J* = 8.1 Hz), 7.56–7.63 (m, 2H, ArH), 7.98 (d, 2H, ArH, *J* = 8.3 Hz), 9.75 (s, 2H, 2NH), 10.63 (s, 1H, NH). ^13^C NMR (DMSO-*d*_6_) δ (ppm): 55.57, 111.00, 112.11, 118.55, 128.84, 130.30, 131.81, 137.18, 140.78, 159.40, 165.54, 181.35. Elemental analysis for C_15_H_14_ClN_3_O_2_S. Calculated: C 53.65; H 4.20; N 12.51. Found: C 53.62; H 4.20; N 12.48.

1-(4-Fluorobenzoyl)-4-(3-methoxyphenyl)thiosemicarbazide (SC2)

CAS 901362-15-0

Yield 59% (0.23 g), m.p. 167–172 °C. Spectral data were as follows: IR (cm^−1^) KBr: 3171 (NH), 3003 (CH aliph.), 1669 (C=O), 1602 (CH arom.), 1363 (C=S), 1280 (C-O-C). ^1^H NMR (DMSO-*d*_6_) δ (ppm): 3.74 (s, 3H, CH_3_), 6.75 (d, 1H, ArH, *J* = 8.2 Hz), 7.04 (d, 1H, ArH, *J* = 8.0 Hz), 7.12 (bs, 1H, ArH), 7.23 (t, 1H, ArH, *J* = 8.1 Hz), 7.31–7.41 (m, 2H, ArH), 7.98–8.05 (m, 2H, ArH), 9.73 (s, 2H, 2NH), 10.57 (s, 1H, NH). ^13^C NMR (DMSO-*d*_6_) δ (ppm): 55.57, 110.97, 112.08, 115.63, 115.78, 118.52, 129.15, 129.52, 131.10, 140.79, 159.40, 163.89, 165.50. Elemental analysis for C_15_H_14_FN_3_O_2_S. Calculated: C 56.41; H 4.42; N 13.16. Found: C 56.41; H 4.37; N 13.14.

4-(3-Methoxyphenyl)-1-(2-trifluoromethylbenzoyl)thiosemicarbazide (SC3)

Yield 46% (0.20 g), m.p. 175–178 °C. Spectral data were as follows: IR (cm^−1^) KBr: 3148 (NH), 2960 (CH aliph.), 1674 (C=O), 1595 (CH arom.), 1335 (C=S), 1265 (C-O-C). ^1^H NMR (DMSO-*d*_6_) δ (ppm): 3.76 (s, 3H, CH_3_), 6.76 (s, 1H, ArH), 7.07 (s, 1H, ArH), 7.27 (s, 2H, ArH), 7.69–7.88 (m, 3H, ArH), 7.98 (s, 1H, ArH), 9.89 (s, 2H, 2NH), 10.57 (s, 1H, NH). ^13^C NMR (DMSO-*d*_6_) δ (ppm): 55.57, 110.76, 121.35, 123.17, 124.98, 126.94, 127.05, 127.27, 129.60, 129.99, 131.12, 132.78, 134.22, 140.68, 140.68, 159.65, 166.45. Elemental analysis for C_16_H_14_F_3_N_3_O_2_S. Calculated: C 52.03; H 3.82; N 11.38. Found: C 51.98; H 3.76; N 11.29.

1-(4-Methoxybenzoyl)-4-(3-methoxyphenyl)thiosemicarbazide (SC4)

CAS 216502-05-5

Yield 56% (0.22 g), m.p. 170–174 °C. Spectral data were as follows: IR (cm^−1^) KBr: 3177 (NH), 2970 (CH aliph.), 1664 (C=O), 1545 (CH arom.), 1355 (C=S), 1258 (C-O-C). ^1^H NMR (DMSO-*d*_6_) δ (ppm): 3.74 (s, 3H, CH_3_), 3.83 (s, 3H, CH_3_), 6.74 (d, 1H, ArH, *J* = 8.2 Hz), 7.04–7.07 (m, 3H, ArH), 7.14 (bs, 1H, ArH), 7.22–7.24 (m, 1H, ArH), 7.94 (d, 2H, ArH, *J* = 8.3 Hz), 9.73 (s, 2H, 2NH), 10.39 (s, 1H, NH). ^13^C NMR (DMSO-*d*_6_) δ (ppm): 55.56, 55.90, 110.88, 111.99, 113.95, 118.46, 125.17, 129.07, 129.19, 130.32, 140.87, 159.36, 162.55, 166.06, 181.39. Elemental analysis for C_16_H_17_N_3_O_3_S. Calculated: C 57.99; H 5.17; N 12.68. Found: C 57.88; H 5.12; N 12.61.

1-(3-Chlorobenzoyl)-4-(3-methoxyphenyl)thiosemicarbazide (SC5)

CAS 891377-64-3

Yield 99% (0,39 g), m.p. 173–178 °C. Spectral data were as follows: IR (cm^−1^) KBr: 3176 (NH), 2959 (CH aliph.), 1667 (C=O), 1596 (CH arom.), 1359 (C=S), 1264 (C-O-C). ^1^H NMR (DMSO-*d*_6_) δ (ppm): 3.75 (s, 3H, CH_3_), 6.75 (d, 1H, ArH, *J* = 8.3 Hz), 7.03 (d, 1H, ArH, *J* = 8.0 Hz), 7.11 (bs, 1H, ArH), 7.24 (t, 1H, ArH, *J* = 8.1 Hz), 7.56 (t, 1H, ArH, *J* = 7.9 Hz), 7.67 (d, 1H, ArH, *J* = 8.0 Hz), 7.90 (d, 1H, ArH, *J* = 7.8 Hz), 8.02 (s, 1H, ArH), 9.77 (s, 2H, 2NH), 10.67 (s, 1H, NH). ^13^C NMR (DMSO-*d*_6_) δ (ppm): 55.58, 111.05, 112.14, 118.66, 127.09, 128.22, 129.19, 130.80, 132.14, 133.55, 135.03, 140.76, 159.42, 165.22, 181.23. Elemental analysis for C_15_H_14_ClN_3_O_2_S. Calculated: C 53.65; H 4.20; N 12.51. Found: C 53.50; H 4.13; N 12.44.

1-(3-Methoxybenzoyl)-4-(4-trifluoromethylphenyl)thiosemicarbazide (SC6)

Yield 65% (0.24 g), m.p. 165–170 °C. Spectral data were as follows: IR (cm^−1^) KBr: 3191 (NH), 2944 (CH aliph.), 1638 (C=O), 1549 (CH arom.), 1327 (C=S), 1273 (C-O-C). ^1^H NMR (DMSO-*d*_6_) δ (ppm): 3.75 (s, 3H, CH_3_), 6.76 (d, 1H, ArH, *J* = 8.3 Hz), 7.04 (d, 1H, ArH, *J* = 8.0 Hz), 7.12 (bs, 1H, ArH), 7.25 (t, 1H, ArH, *J* = 8.1 Hz), 7.91 (d, 2H, ArH, *J* = 8.2 Hz), 8.16 (d, 1H, ArH, *J* = 8.0 Hz), 9.81 (s, 2H, 2NH), 10.80 (s, 1H, NH). ^13^C NMR (DMSO-*d*_6_) δ (ppm): 55.57, 111.05, 112.16, 118.26, 121.66, 123.46, 125.27, 125.75, 129.28, 131.82, 132.03, 132.24, 132.45, 136.88, 140.74, 159.45, 165.40. Elemental analysis for C_16_H_14_F_3_N_3_O_2_S. Calculated: C 52.03; H 3.82; N 11.38. Found: C 51.99; H 3.78; N 11.37.

Synthesis of 1,3,4-thiadiazoles

Synthesis of SCT1 and SCT2

First, 0.2 g of the thiosemicarbazide derivatives obtained in the earlier stage was weighed and placed in conical flasks. Then, 0.25 mL of concentrated sulfuric acid(VI) was added dropwise with thorough stirring. The reaction mixture was stirred until the precipitate dissolved, and 2 h were given for cooling at room temperature. Then, finely crushed ice was added to the reaction flasks and mixed intensively, until solid completely precipitated. After the complete dissolution of the ice, the solid product was filtered off. The product was dried carefully with filter paper. Then, crystallization from 2-propanol was performed to obtain pure SCT1 and SCT2 compounds.

Synthesis of SCT4, SCT5, and SCT6

To 0.2 g of the thiosemicarbazide derivatives, 0.25 mL of concentrated sulfuric acid(VI) was added dropwise. The reaction mixture was stirred for 0.5 h at room temperature. Then, crushed ice was added to the reaction flasks and mixed intensively, until the solid completely precipitated. The precipitate formed which was allowed to dissolve the ice at room temperature. The solid product was filtered off, dried thoroughly with filter paper, and crystallized from butanol to obtain pure SCT4, SCT5 and SCT6 compounds.

Synthesis of SCT3

We tried to synthesize SCT3 from thiosemicarbazide derivatives SC3 by modifying the laboratory method in various ways. We run the reactions in the two ways described above, changing the time of exposure to concentrated sulfuric(VI) acid and the type of solvent used. We also tried using a microwave at 30 °C for 10 min to perform the synthesis. Changing the reaction conditions using microwaves also did not bring the expected result. Unfortunately, in each modification variant of the laboratory method, the cyclization of SC3 to SCT3 did not take place, which was confirmed using the TLC and ^1^H NMR methods.

5-(4-Chlorophenyl)-2-(3-methoxyphenylamino)-1,3,4-thiadiazole (SCT1) [33]

CAS 143722-18-3

Yield 45% (0.085 g), m.p. 210–214 °C. Spectral data were as follows: IR (cm^−1^) KBr: 3257 (NH), 2834 (CH aliph.), 1622 (C=N), 1572 (CH arom.), 1260 (C-O-C), 741 (C-S). ^1^H NMR (DMSO-*d*_6_) δ (ppm): 3.77 (s, 3H, CH_3_), 6.62 (dd, 1H, ArH, *J* = 8.1 Hz, *J* = 2.5 Hz), 7.14 (dd, 1H, ArH, *J* = 8.1 Hz, *J* = 2.1 Hz), 7.27 (t, 1H, ArH, *J* = 8.1 Hz), 7.38 (t, 1H, ArH, *J* = 2.3 Hz), 7.57 (d, 2H, ArH, *J* = 8.6 Hz), 7.88 (d, 2H, ArH, *J* = 8.6 Hz), 10.60 (s, 1H, NH). ^13^C NMR (DMSO-*d*_6_) δ (ppm): 55.52, 104.03, 108.05, 110.46, 128.86, 129.62, 129.76, 130.41, 135.17, 142.02, 156.88, 160.44, 164.79. Elemental analysis for C_15_H_12_ClN_3_OS. Calculated: C 56.69; H 3.81; N 13.22. Found: C 56.61; H 3.62; N 13.19.

5-(4-Fluorophenyl)-2-(3-methoxyphenylamino)-1,3,4-thiadiazole (SCT2)

Yield 53% (0.10 g), m.p. 207–209 °C. Spectral data were as follows: IR (cm^−1^) KBr: 3204 (NH), 2836 (CH aliph.), 1620 (C=N), 1575 (CH arom.), 1250 (C-O-C), 745 (C-S). ^1^H NMR (DMSO-*d*_6_) δ (ppm): 3.77 (s, 3H, CH_3_), 6.62 (dd, 1H, ArH, *J* = 8.2 Hz, *J* = 2.5 Hz), 7.13 (dd, 1H, ArH, *J* = 8.1 Hz, *J* = 2.3 Hz), 7.27 (t, 1H, ArH, *J* = 8.1 Hz), 7.35–7.39 (m, 3H, ArH), 7.91–7.94 (m, 2H, ArH), 10.56 (s, 1H, NH). ^13^C NMR (DMSO-*d*_6_) δ (ppm): 55.52, 103.96, 107.98, 110.41, 116.79 (d, *J* = 22.0 Hz), 127.38 (d, *J* = 3.3 Hz), 129.51 (d, *J* = 8.6 Hz), 130.40, 142.08, 156.98, 160.44, 162.72, 164.36, 164.59. Elemental analysis for C_15_H_12_FN_3_OS. Calculated: C 59.79; H 4.01; N 13.95. Found: C 59.68; H 3.97; N 13.67.

5-(4-Methoxyphenyl)-2-(3-methoxyphenylamino)-1,3,4-thiadiazole (SCT4) [33]

CAS 143722-16-1

Yield 32% (0.09 g), m.p. 189–190 °C. Spectral data were as follows: IR (cm^−1^) KBr: 3201 (NH), 2838 (CH aliph.), 1624 (C=N), 1578 (CH arom.), 1249 (C-O-C), 744 (C-S). ^1^H NMR (DMSO-*d*_6_) δ (ppm):): 3.77 (s, 3H, CH_3_), 3.82 (s, 3H, CH_3_), 6.60 (d, 1H, ArH, *J* = 8.0 Hz) 7.06 (d, 2H, ArH, *J* = 8.4 Hz), 7.13 (d, 1H, ArH, *J* = 8.1 Hz), 7.26 (t, 1H, ArH, *J* = 8.1 Hz), 7.38 (s, 1H, ArH), 7.80 (d, 2H, ArH, *J* = 8.3 Hz), 10.47 (s, 1H, NH). ^13^C NMR (DMSO-*d*_6_) δ (ppm): 55.50, 55.86, 103.88, 107.82 110.33, 115.11, 123.33, 128.80, 130.36, 142.23, 157.96, 160.44, 161.27, 163.86. Elemental analysis for C_16_H_15_N_3_O_2_S. Calculated: C 61.32; H 4.82; N 13.41. Found: C 61.28; H 4.78; N 13.35.

5-(3-Chlorophenyl)-2-(3-methoxyphenylamino)-1,3,4-thiadiazole (SCT5)

Yield 53% (0.10 g), m.p. 195–197 °C. Spectral data were as follows: IR (cm^−1^) KBr: 3204 (NH), 2837 (CH aliph.), 1624 (C=N), 1581 (CH arom.), 1251 (C-O-C), 738 (C-S). ^1^H NMR (DMSO-*d*_6_) δ (ppm): 3.78 (s, 3H, CH_3_), 6.63 (d, 1H, ArH, *J* = 8.2 Hz), 7.14 (d, 1H, ArH, *J* = 8.1 Hz), 7.26–7.29 (m, 1H, ArH), 7.38 (s, 1H, ArH), 7.55 (s, 2H, ArH), 7.82 (d, 1H, ArH, *J* = 8.2 Hz), 7.92 (s, 1H, ArH), 10.64 (s, 1H, NH). ^13^C NMR (DMSO-*d*_6_) δ (ppm): 55.54, 104.06, 108.16, 110.50, 126.04, 126.43, 130.44, 131.67, 132.70, 134.40, 141.97, 156.53, 160.46, 165.06. Elemental analysis for C_15_H_12_ClN_3_OS. Calculated: C 56.69; H 3.81; N 13.22. Found: C 56.68; H 3.78; N 13.19.

2-(3-Methoxyphenylamino)-5-(4-trifluorometylophenyl)-1,3,4-thiadiazole (SCT6)

Yield 49% (0.09 g), m.p. 230–233 °C. Spectral data were as follows: IR (cm^−1^) KBr: 3207 (NH), 2840 (CH aliph.), 1624 (C=N), 1577 (CH arom.), 1253 (C-O-C), 744 (C-S). ^1^H NMR (DMSO-*d*_6_) δ (ppm): 3.78 (s, 3H, CH_3_), 6.64 (d, 1H, ArH, *J* = 8.1 Hz), 7.15 (d, 1H, ArH, *J* = 8.1 Hz), 7.28 (t, 1H, ArH, *J* = 8.1 Hz), 7.39 (s, 1H, ArH), 7.87 (d, 2H, ArH, *J* = 8.0 Hz), 8.09 (d, 2H, ArH, *J* = 8.0 Hz), 10.69 (s, 1H, NH). ^13^C NMR (DMSO-*d*_6_) δ (ppm): 55.54, 104.16, 108.23, 110.57, 123.52, 125.32, 126.63, 127.87, 130.45, 134.50, 141.91, 156.51, 160.46, 165.36. Elemental analysis for C_16_H_12_N_3_F_3_OS. Calculated: C 54.70; H 3.44; N 11.96. Found: C 54.67; H 3.46; N 11.99.

### 3.2. Cell Culture 

Cell culture MCF-7, MDA-MB-231 human breast cancer cells, and fibroblasts skin cells were purchased from the ATCC—American Type Culture Collection. All cell lines were maintained in DMEM (Corning, Kennebunk, ME, USA). The medium was supplemented with 10% fetal bovine serum—FBS (Eurx, Gdansk, Poland) and 1% antimicrobial substances: penicillin–streptomycin (Corning, Kennebunk, ME, USA). The incubator established appropriate growth conditions required for these cell lines: 5% of CO_2_, at 37 °C, with the humidity between 90 and 95%. Cells were seeded in 100 mm round dishes. An amount of 0.05% trypsin containing 0.02% EDTA (Corning, Kennebunk, ME, USA) and phosphate-buffered saline (PBS) without calcium and magnesium (Corning, Kennebunk, ME, USA) was used to detach cells from the plate once 80–90% cell confluence was achieved. In the next step, cells were reseeded in six-well plates (density 5 × 10^5^ of cells per well) in 1 mL of DMEM after a 24 h incubation used in the presented tests.

### 3.3. Cell Viability Assay 

To examine the effect of the tested compounds (SCT1, SCT2, SCT4, SCT5 and SCT6) on cell viability, the MTT assay was performed. Cells, seeded in six-well plates, were incubated for 24 h with serial dilutions of the tested compounds in duplicates. In the next step, the liquid was aspirated above the cells and cells were washed three times with PBS. Thereafter, 50 µL of 5 mg/mL of MTT (Sigma Aldrich, St Louis, MO, USA) was added to 1 mL of PBS. After the required time, the MTT solution was removed and resulting formazan crystals were dissolved in DMSO. The absorbance was measured using Spectrophotometer UV-VIS Helios Gamma (Unicam/ThermoFisher Scientific Inc., Waltham, MA, USA) at a wavelength of 570 nm. The obtained absorbance in untreated control cells was taken as 100%, while the survival of the cells incubated with tested compounds was presented as a percentage of the control value [34].

### 3.4. [^3^H]-Thymidine Incorporation Assay

The antiproliferative properties of the newly synthesized compounds were investigated through the [^3^H]-thymidine incorporation assay. Cell cultures were exposed to various concentrations of drugs for 24 h. Thereafter, cells were washed with PBS and 1 mL of fresh medium was added to each well. Then, 0.5 µCi of radioactive [^3^H]-thymidine was added and the incubation continued for four hours. After the following incubation, the liquid was aspirated and the plate was placed on ice. Cells were washed twice with 1 mL of 0.05 M Tris-HCl buffer comprising 0.11 M NaCl, then twice with 1 mL of 5% TCA acid (Stanlab, Lublin, Poland). Finally, the cells were dissolved with 1 mL of 0.1 M NaOH with 1% SDS (Sigma Aldrich, St. Louis, MO, USA) at room temperature (RT). The resulting cell lysates were transferred into scintillation vials containing 2 mL of scintillation fluid. The radioactivity was determined using Scintillation Counter 1900 TR, TRI-CARB (Packard, Perkin Elmer, Inc., San Jose, CA, USA). The intensity of DNA biosynthesis in cells was expressed in dpm of radioactive thymidine incorporated in the DNA. The radioactivity observed in untreated control cells was taken as 100%. Values from the tested compounds were expressed as a percentage of the control value [35].

### 3.5. Docking Simulations

All the docking techniques, software, and validation procedures were the same as in the previous article [26].

### 3.6. Caspase 3/7 and 8 Activity Assay

Activation of the caspase cascade occurs as a result of the initiation of the apoptotic process in the cell and is induced by the cytotoxic activity of the compound. In this regard, assessment of initiator (caspase 8) and executioner (caspase 3 and 7) caspases activity was performed using FAM-FLICA^®^ Caspase Assays kits (all from ImmunoChemistry Technologies, Bloomington, MN, USA) according to the manufacturer’s instructions. After 24 h incubation of MCF-7 and MDA-MB-231 breast cancer cells with the tested compounds (SCT4 and SCT5) at concentrations of 100 µM, cells were collected, washed twice with cold PBS, and resuspended in Apoptosis Wash Buffer to a final concentration of 5 × 10^5^ cells/mL. In the next step, 290 µL each of cell suspensions was taken and transferred into tubes. Then, 10 µL each of FLICA solution diluted immediately before use (1:5 *v*/*v*, using PBS) was added to the cells, mixed by pipetting, and incubated in the dark for 1 h at 37 °C. After this time, cells were washed twice with 2 mL Apoptosis Wash Buffer, centrifuged, and resuspended in 300 µL of the buffer. Thus, prepared samples were immediately analyzed using a BD FACSCanto II flow cytometer (10,000 events) with FACSDiva software (both from BD Biosciences Systems, San Jose, CA, USA). The equipment calibration was performed using BD Cytometer Setup and Tracking Beads (BD Biosciences, San Diego, CA, USA).

### 3.7. Flow Cytometry Assessment of Annexin V and Propidium Iodide Binding

The apoptosis induction by the tested compounds was assessed by the exposure of phosphatidylserine on the cell membrane, to which fluorescein isothiocyanate (FITC)-labeled Annexin V binds with high affinity in the presence of Ca^2+^ calcium ions. The FITC Annexin V Apoptosis Detection Kit II (BD Pharmingen, San Diego, CA, USA) and a flow cytometer (BD FACSCanto II, BD Biosciences Systems, San Jose, CA, USA) were used for this detection. The assay was performed according to the manufacturer’s instructions. Breast cancer cells (MCF-7 and MDA-MB-231) were incubated for 24 h (37 °C, 5% CO_2_, 90–95% humidity) with the most active compound (SCT4 and SCT5) at concentrations of 100 µM. Flow cytometer calibration was performed by preparing two controls—a positive and negative control. The positive control were cells in which apoptosis was induced using 2 µL of 3% formaldehyde in buffer and placing them on ice for 30 min. In contrast, the negative control was represented by cells that were not treated with any of the proapoptotic agents. First, in cells treated with the tested compounds as well as the controls, the medium was removed and the cells were washed twice with cold PBS. Subsequently, cells were resuspended in Binding Buffer included in the kit at a concentration of 1 × 10^6^ cells/mL. From each sample, 100 µL of cell suspension was taken and transferred to test tubes to which 5 µL each of FITC Annexin V and propidium iodide (PI) were then added. The contents of the test tubes were gently vortexed and incubated for 15 min at room temperature, protected from light. After the required time, the contents of the test tubes were made up to 500 µL with Binding Buffer and immediately analyzed in a flow cytometer (10,000 events measured). After flow cytometer readout, results were analyzed using FACSDiva software (BD Biosciences Systems, San Jose, CA, USA). The equipment was calibrated with BD Cytometer Setup and Tracking Beads (BD Biosciences, San Diego, CA, USA).

## 4. Conclusions

New derivatives of thiosemicarbazide and 1,3,4-thiadiazole were synthesized and evaluated for their in vitro anticancer activity. The results showed that all compounds showed weak anticancer activity towards two breast cancer lines: an estrogen-dependent cell line (MCF-7) and an estrogen-independent cell line (MDA-MB-231). Docking simulation revealed the decreased binding energy for the tested compounds to the analyzed proteins. In particular, we observed binding energy decreases in caspase 8–SCT complexes, which confirms the theory about the relationship between the cytotoxic activity of the synthesized compounds and its influence on caspase 8.

Comparing the results obtained in the previous work with the results of the presented research it can be concluded that activity in this class of compounds is determined by the substituent derived from the starting hydrazide not by isothiocyanate used for synthesis. The collected data indicate a significant influence of the presence of methoxy groups in the *meta* position located next to the phenyl ring in position 2 in relation to the thiadiazole heterocyclic system. This shows us a further direction in the search for new, more active thiadiazole derivatives, e.g., among derivatives with dimethoxyphenyl and trimethoxyphenyl groups in position 2.

## Data Availability

The details of the data supporting the report results in this research were included in this paper and Appendix A.

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
