# Peer review of "Synthesis and Anticancer Activity of 1,3,4-Thiadiazoles with 3-Methoxyphenyl Substituent"

_molecules, 2022, doi:10.3390/molecules27206977_

Round 1

Reviewer 1 Report

The authors have employed a wide range of concentrations without calculating the IC50, making their biological activity results unacceptable in their current form. Its not authentic and logical as it also demonstrates that compounds are not considerably active.

Author Response

We agree with the Reviewer that we obtained the 1,3,4-thiadiazole derivatives with weak anticancer activity. IC50 was higher than 100 µM. The most effective derivative was SCT-4, which decreased DNA biosynthesis to 70.46% ± 3 at 100 µM. The results are authentic and we would like to underline that we put a lot of effort into carrying out all experiments (design, synthesis, biological assays, and docking simulations) in the context of their anticancer activity. We decided to show the results to the other researchers despite the weak anticancer potential of the tested compounds.

In our opinion, the presentation of our results to the reader is necessary to prove that such chemical structures are not very promising and effective in in vitro conditions. We believe that the presented results are worthy of attention and this manuscript is an example how hard work we have to put in to find novel, very active anti-breast cancer molecules.

On the basis of the obtained results, it can be stated that the presence of methoxy groups in the meta position located on the phenyl ring in the 2-position in relation to the heterocyclic system of thiadiazole is significant. This shows us a further direction in the search for new, more active 1,3,4-thiadiazole derivatives, including derivatives with more than one methoxy group in the phenyl ring at the 2-position.

The manuscript has been proofread by a native speaker.

Changes are marked in red.

We would like to thank the Reviewer for the opportunity to strengthen the manuscript. We hope that the revised version meets your approval.

Reviewer 2 Report

The manuscript molecules-1930283, Synthesis and anticancer activity of 1,3,4-thiadiazoles with 3-methoxyphenyl substituent, describes the synthesis of 13 derivatives carrying an amino thiadiazole scaffold decorated with substituted aromatic moieties. These compounds are claimed to have an antitumor activity, with a suggested mechanism of action involving caspase 8.

The paper reports many biological and docking studies but, unfortunately, the number of compounds is small, and the derivatives have few structural differences; moreover, these derivatives do not have a significant effect. The authors themselves state that the compounds are not cytotoxic to tumor cell lines. Most notably, the research strategy underlying this study is unclear and the exact aim (what information and which mechanism are the authors looking for?) is not well defined.

In my opinion, in the present form, the paper cannot be accepted for publication on Molecules, but must be expanded with other derivatives, to obtain information able to outline sound SAR, and must be completely reorganized.

In the forthcoming paper, the author should consider the following remarks:

-some statements (e.g.: this heterocyclic system is a bioisoster of pyrimidine…., or: the therapeutic potential of thiadiazole molecules….) need references;

- an improved quality of the designed molecules should be provided;

- the font dimension should be checked throughout the manuscript;

-the viability data reported in Table 1 and in Figures 1-3 seem to be different; are they referred to the same experiment lasting 24 h? Data regarding the second series are not reported in the Table but only in the Figures. Therefore, the discussion is quite confusing.

- in the conclusion section, some statements are really unclear.

I also suggest a careful revision by a native-speaking proof-reader.

Author Response

The presented manuscript is a continuation of our research work presented in the previous issue of Molecules (reference 26). In the previous paper, we proved that of 1,3,4-thiadiazoles derivatives possess promising anticancer properties against the MDA-MB-231 and MCF-7 cell lines [26]. The new series is a modification of the structures of the most active compounds from the previous work [26]. To check whether the activity of 1,3,4-thiadiazole is more influenced by the substituent attached to the ring with the mobile amino group, which allows its movement, or by the substituent rigidly attached directly to the heterocyclic ring, we swapped the substituents in the new molecules. The viability of cancer cells was checked by MTT assay and anti-proliferative effect was tested by DNA biosynthesis. The pro-apoptotic potential of the novel derivative was confirmed by flow cytometry. The activity of caspases, which are engaged in the extrinsic and intrinsic apoptotic pathway was determined.

We agree with the Reviewer that we obtained the 1,3,4-thiadiazole derivatives with weak anticancer activity. IC50 was higher than 100µM. The most effective derivative was SCT-4, which decreased DNA biosynthesis to 70,46% ± 3 at 100 µM. The results are authentic and we would like to underline that we put a lot of efforts into carrying out all the experiments (design, synthesis, biological assays and docking simulations) in the context of their anticancer activity. We decided to show the results to the other researchers despite the weak anticancer potential of the tested compounds. In our opinion, the presentation of our results to readers is necessary to prove that such chemical modification of 1,3,4-thiadiazoles are not very promising and effective in vitro conditions. We believe that the presented results are worthy of attention.

The obtained results will be used to rationally design further compounds with potential anti-cancer activity. We suppose that introducing further methoxy groups into the phenyl ring in position 2 will lead to an increase in anti-tumor activity. This will be the subject of our further research.

The manuscript has been proofread by a native speaker.

Changes are marked in red.

We would like to thank the Reviewer for the opportunity to strengthen the manuscript. We hope that the revised version meets your approval.

Reviewer 3 Report

In my opinion, the article could be published as it is, however there are a few small issues which authors have to address.   1)      Data in Table 1 and 2 are given with two significant digits after decimal point while standard errors are whole number, to me it doesn’t make any sense from statistical point of view   2)      In the spectral data of the SC5 compound (p.16) the IR value 2002 cm-1 for CH aliphatic bond is obviously wrong/mistaken   3)      In general, one is not supposed to cut out NMR spectra like the  authors did in supplementary material with proton NMR  range of 1.5-11 ppm. The full scale (0-12 ppm) has to be presented to ensure absence of aliphatic impurities. Also as a rule of thumb, height/intensity of the smallest peak in 1H NMR spectrum should be roughly 25-30% of page height.   4)      There are no integrals in a spectrum of SCT4 (Fig. S9, Supplementary materials)

Author Response

Detailed comments regarding the manuscript were corrected in line with the Reviewer's comments. According to the suggestions:

  1. Data in Table 1 and 2 are given with two significant digits after decimal point while standard errors are whole number, to me it doesn’t make any sense from statistical point of view

According to the suggestion, data in Table 1 and 2 were corrected.

  1. In the spectral data of the SC5 compound (p.16) the IR value 2002 cm-1 for CH aliphatic bond is obviously wrong/mistaken.

The value was corrected.

  1. In general, one is not supposed to cut out NMR spectra like the  authors did in supplementary material with proton NMR  range of 1.5-11 ppm. The full scale (0-12 ppm) has to be presented to ensure absence of aliphatic impurities. Also as a rule of thumb, height/intensity of the smallest peak in 1H NMR spectrum should be roughly 25-30% of page height.

NMR spectra were corrected. The concentration of the compound in the sample, due to poor solubility, did not allow obtaining the proper spectrum with the appropriate peak height.

  1. There are no integrals in a spectrum of SCT4 (Fig. S9, Supplementary materials)

NMR spectra was corrected.

Changes are marked in red.

We would like to thank the Reviewer for the opportunity to strengthen the manuscript. We hope that the revised version meets your approval.

Round 2

Reviewer 1 Report

As authors stated in response that" we put a lot of effort into carrying out all experiments (design, synthesis, biological assays, and docking simulations) in the context of their anticancer activity. We decided to show the results to the other researchers despite the weak anticancer potential of the tested compounds"
So I agree to accept the manuscript.

Reviewer 2 Report

I have carefully read the revised work and the authors' response to my comments, and absolutely I do not question the great work done from the point of view of the biological characterization of these compounds. I also agree that even unsatisfactory data can be used for a future rational design of new anticancer compounds. However, my opinion has not changed regarding the manuscript, which was only slightly modified. I just want to point out a few highlights.

The rationale behind this project and in particular the rationale that led to the synthesis of these new derivatives is really not well explained, as I pointed out in more detail in my previous comment. In any case, these few (6) products carry only a small modification compared to the major number of compounds (15 + 15) described in the previous work released in March 2022. Unfortunately, this modification is not productive, because the new compounds are very little active; moreover, of 6 new compounds only 5 were tested. It should be more clearly presented which compounds have been obtained previously and which are original, and the comparison with the preceding biological data is not clearly discussed. In any case, also the previously obtained compounds that are reported in the present paper do not show a high activity. I think this work can be turned into a manuscript of biological data only and sent to a dedicated journal; otherwise it is necessary to expand it to be more informative from a chemical-pharmaceutical point of view (see previous revision).